# Exposure to 3′Sialyllactose-Poor Milk during Lactation Impairs Cognitive Capabilities in Adulthood

**DOI:** 10.3390/nu13124191

**Published:** 2021-11-23

**Authors:** Edoardo Pisa, Alberto Martire, Valentina Chiodi, Alice Traversa, Viviana Caputo, Jonas Hauser, Simone Macrì

**Affiliations:** 1Centre for Behavioural Sciences and Mental Health, Istituto Superiore di Sanità, 00161 Rome, Italy; edoardo.pisa@iss.it; 2Department of Physiology and Pharmacology, Sapienza University of Rome, 00185 Rome, Italy; 3National Centre for Drug Research and Evaluation, Istituto Superiore di Sanità, 00161 Rome, Italy; alberto.martire@iss.it (A.M.); valentina.chiodi@iss.it (V.C.); 4Laboratory of Clinical Genomics, Fondazione IRCCS Casa Sollievo della Sofferenza, 71013 San Giovanni Rotondo, Italy; a.traversa@css-mendel.it; 5Department of Experimental Medicine, Sapienza University of Rome, 00161 Rome, Italy; viviana.caputo@uniroma1.it; 6Brain Health, Nestlé Institute of Health Science, Nestlé Research, Société des Produits Nestlé SA, 1000 Lausanne, Switzerland

**Keywords:** breast milk, executive function, memory, brain, lactation, 3′-sialyllactose, human milk oligosaccharides, sialic acid

## Abstract

Breast milk exerts pivotal regulatory functions early in development whereby it contributes to the maturation of brain and associated cognitive functions. However, the specific components of maternal milk mediating this process have remained elusive. Sialylated human milk oligosaccharides (HMOs) represent likely candidates since they constitute the principal neonatal dietary source of sialic acid, which is crucial for brain development and neuronal patterning. We hypothesize that the selective neonatal lactational deprivation of a specific sialylated HMOs, sialyl(alpha2,3)lactose (3′SL), may impair cognitive capabilities (attention, cognitive flexibility, and memory) in adulthood in a preclinical model. To operationalize this hypothesis, we cross-fostered wild-type (WT) mouse pups to B6.129-*St3gal4*^tm1.1Jxm^/J dams, knock-out (KO) for the gene synthesizing 3′SL, thereby providing milk with approximately 80% 3′SL content reduction. We thus exposed lactating WT pups to a selective reduction of 3′SL and investigated multiple cognitive domains (including memory and attention) in adulthood. Furthermore, to account for the underlying electrophysiological correlates, we investigated hippocampal long-term potentiation (LTP). Neonatal access to 3′SL-poor milk resulted in decreased attention, spatial and working memory, and altered LTP compared to the control group. These results support the hypothesis that early-life dietary sialylated HMOs exert a long-lasting role in the development of cognitive functions.

## 1. Introduction

Among the numerous factors modulating development, nutrition exerts a paramount role since the very early stages of life [1,2,3]. Breast milk constitutes the main source of nutrition for a newborn whereby it is involved in the development of numerous structures and biological systems [4,5]. Its composition is a result of evolutionary adaptive processes through which the milk of various mammalian species evolved to best suit the needs of that given species in terms of the relative proportions of its different constituents [6]. For example, in several arid-adapted mammalian species, the milk is relatively diluted, facilitating the evaporative cooling of the offspring [7].

With respect to its adaptive function in ontogeny, maternal milk contains several nutrients implicated in the development of, for example, the immune system [8,9] and the gastrointestinal tract [10,11,12]. Recently, it has been demonstrated that the regulatory role of breast milk extends well beyond physiological functions to involve the development of brain structures and the associated cognitive functions [11]. The benefits exerted by maternal milk have been substantiated by retrospective studies comparing the phenotypes of individuals reared to breastfeeding (BF) or formula-feeding (FF). Thus, children who have not been breastfed or breastfed for a short period have been reported to show a lower verbal IQ score and poorer school performances than children who have been breastfed for a period of 6 months or more [5]. These psychometric data have been supported by brain imaging studies showing that adolescent boys, exclusively exposed to breast milk during lactation, exhibited higher white matter growth compared to subjects that, during lactation, were exposed to a mixed diet [13]. These data, along with additional experimental evidence (e.g., [4,5,14]), contribute to the World Health Organization recommendation to exclusively breastfeed infants for the first 6 months of life [15].

Among the several nutrients present in human milk, sialic acid (Sia, a family of 9-carbon sugar acids) has been proposed to constitute a key nutrient for neurodevelopment during the early postnatal stages [8], whereby it acts as a building block for gangliosides and poly-sialylated neural cell adhesion molecules (Poly-Sia-NCAMs). Likewise, sialic acid serves as a binding site for myelin-associated glycoprotein (MAG), modulating the interaction between nerve and glial cells [16]. Gangliosides are glycosphingolipids that contain at least one acidic Sia residue, representing about 6 to 10% of the total lipids in the human brain. The concentration of brain gangliosides increases during the first two years of life, confirming the crucial role exerted by sialic acid during the early stages of development [17,18]. Recently, brain ganglioside expression has been correlated with some neurophysiological functions, such as neurogenesis, synaptogenesis, and memory formation [19]. To this direction, the lack of brain gangliosides leads to axon degeneration and disrupted myelination both in the central and peripheral nervous systems [17,20,21].

In mammalian cells, Sia monomers can polymerize into linear anionic chains of sialyl residues (Poly-Sia) [22]. In human neural cells, these structures post-translationally modify and regulate the function of NCAMs during development [23], including cell migration, neurite outgrowth, pathfinding, sprouting, and regeneration in hippocampus [18]. *N*-acetylneuraminic acid (2-keto-5-acetamido-3,5-dideoxy-d-glycero-d-galactononulosonicacid) is the most common form of Sia in human glycoconjugates. Importantly, while adult mammals can endogenously synthesize Sia from glucose and other products of glycolysis, newborns lack this capability, requiring an exogenous source of Sia [24]. Therefore, maternal milk has been proposed to represent a fundamental exogenous source of Sia to developing offspring [18]. Such a pivotal role is attained through sialylated human milk oligosaccharides (HMOs), which represent one of the main dietary sources of Sia to the lactating offspring. Additional evidence in favor of the role of HMOs in the development of cognitive capabilities stems from the observation that genetic polymorphisms leading to variations in HMO concentrations relate to differential development of the receptive and expressive domains of language in children [25]. The evolutionary preserved role of HMOs is highlighted by the presence of these constituents in many diverse mammalian species [6,26]. The relative concentration of these constituents has been shown to considerably fluctuate among different species (see [26,27,28]), with donkey milk attaining particularly elevated concentrations of sialylated oligosaccharides compared to other species [29]. 

Besides the molecular mechanisms underlying the role of Sia on brain development and the observational data collected in humans, several preclinical studies highlight the role of Sia on cognitive development. Wang and colleagues exposed three-day-old piglets to a diet supplemented with sialic acid for five weeks and observed improved learning and memory, and increased concentrations of Sia in the frontal cortex at the end of treatment [30]. Recently, Oliveros and colleagues showed that a neonatal exogenous compensatory supplementation of Sia mitigated the long-term negative consequences (on learning and memory) of Sia deprivation during lactation [31]. While these studies highlighted the importance of sialylated oligosaccharides in developing cognitive functions, they nonetheless present some limitations. Specifically, they did not mimic the real-life situation whereby they either supplemented Sia in a condition of physiological availability [30] or in a condition in which the maternal milk received by rat pups was radically different from that naturally received during this specific life stage [31]. In a recently published paper, these limitations were overcome using an experimental design that allowed to study the selective absence of sialyl(alpha2,6)lactose (hereafter 6′SL) only during lactation in mice [32]. Together with sialyl(alpha2,3)lactose (hereafter 3′SL), they are the most abundant sialylated HMOs in the human breast milk [11]. This study leveraged a mouse knock-out (KO) model in which the gene responsible for the synthesis of 6′SL was deleted. This genetic engineering procedure resulted in dams providing milk devoid of 6′SL [33]. To expose wild-type (WT) mice to a milk deprived of 6′SL, a cross-fostering procedure was performed in which WT mice were transferred to a KO dam. The mice were then assessed for their individual cognitive capabilities in adulthood. Compared to the control mice, the WT mice deprived of 6′SL during lactation exhibited impairments in memory, attention, and hippocampal LTP, which is a cellular process thought to be associated with memory [32]. Furthermore, these data were associated to a time- and region-specific reduced expression of genes related to neurogenesis and synaptic plasticity, highlighting the potential regulatory role of 6′SL.

In the present study, we aimed at investigating the role exerted by a specific sialylated HMO, the 3′SL, on the development of higher cognitive functions. We focused on 3′SL due to the fact that 3′SL plays a major role in brain structure pattering, gut microbiota homeostasis, and immunity during development [33] and that 3′SL is abundant in maternal milk but not present in infant formula [11]. Finally, while there are multiple studies investigating the neurodevelopmental role of early-life dietary 6′SL intake, the other major source of sialic acid in rodent milk, literature on the role of 3′SL is scant. Based on these considerations, we tested the hypothesis that a selective reduction of 3′SL during lactation may persistently impair cognitive capabilities in developing subjects. 

To test our hypothesis, we leveraged a mouse model characterized by a constitutive absence of 3′SL (B6.129-*St3gal4*^tm1.1Jxm^, hereafter knock-out, KO) [34]. The inactivated gene, *St3Gal4*, encodes for an α2,3-sialyltransferase, which mediates the synthesis of 3′SL in mouse mammary gland [8]. Previous studies, which quantified the content of 3′SL and 6′SL throughout lactation, reported that the maternal milk of these KO dams has a constantly reduced concentration of 3′SL (approximatively 80% reduction) with no significant variation in the content of 6′SL [8]. By cross-fostering the WT pups to the KO dams, we were able to investigate the effect of a 3′SL-poor diet during lactation on attention, spatial and object memory, and hippocampal LTP in adulthood. To distinguish between the impact of the deletion of *St3gal4* gene and 3′SL conditional dietary intake reduction during lactation, we also evaluated the profile exhibited by KO offspring reared to KO or WT dams. Finally, since our experimental approach required WT pups reared to KO dams and KO pups reared to WT dams, we controlled for cross-fostering procedures by performing in-fostering in WT to WT and KO to KO instances. To avoid any confounder of potential preferential maternal care for own offspring, all mice were reared to foster nonbiological dams. Finally, to minimize the number of subjects used in the experiment, we adopted a split-litter cross-fostering design. Thus, litters were composed of an identical number of WT and KO offspring. 

## 2. Materials and Methods

### 2.1. Animals and Rearing Conditions

Adult wild-type (WT) B6.129 and heterozygous (HZ) B6.129-*St3gal4*^tm1.1Jxm^/J breeding mice pairs (four males and four females and three males and four females, respectively, 50 days of age at arrival) were purchased from a commercial breeder (the Jackson Laboratory, Bar Harbor, ME, USA). Upon arrival, the mice were housed in same-sex and genotype groups of 2–3 in type I polycarbonate cages (33.0 × 13.0 × 14.0 cm) equipped with an enrichment bag (Mucedola, Settimo Milanese, Italy), metal top and ad libitum water, food pellets (Mucedola, Settimo Milanese, Italy), and with the floor covered with sawdust bedding. Housing facilities were maintained on a reversed 12 h light–dark cycle (lights on at 7:00 p.m.) in an air-conditioned room (relative humidity 60 ± 10% and temperature 21 ± 1 °C). Two weeks after arrival, one male and two females of the same genotype were formed. Male mice were removed after two weeks of mating, and females were housed individually in standard cages. The females were checked daily for delivery and the day they gave birth was designated as postnatal day (PND) 0. Dams and their offspring were kept undisturbed until weaning (on PND 28), apart from cage cleaning once a week. At weaning, male and female mice were separated and located in same-sex, same-litter cages; additionally, the male mice were marked by ear clippings. Tips of the tails were collected and used for genotyping. Fourteen homozygous KOs and 14 WT mice were then used for the experiments, while heterozygous mice were euthanized by increased concentrations of CO_2_.

### 2.2. Genotyping Procedure

To extract genomic DNA, 0.3–0.5 cm of mouse tail biopsies were incubated at 55 °C overnight in a heatblock (Thermomixer^®^ R, Eppendorf, Hamburg, Germany) with gentle agitation (300 rpm) in 0.5–0.25 mL of Lysis Buffer, pH 8 (100 mM Tris–HCl, 0.5% *v*/*v* TWEEN^®^ 20, 0.5% *v*/*v* NP-40, pH adjustment with HCl) with 0.1 mg/mL Proteinase K (Thermo Fisher Scientific, Waltham, MA, USA). After lysis, Proteinase K was inactivated by incubating the samples at 75 °C for 20 min. Two microliters of each lysate were used to perform the PCR reaction with GoTaq^®^ G2 Flexi DNA Polymerase (Madison, WI, USA) (1,3U GoTaq^®^ G2 Flexi DNA Polymerase, 1× GoTaq^®^ Flexi Buffer, 1.5 mM MgCl_2_ solution, 0.125 mM each dNTP, 0.6 µM each primer, molecular biology grade water up to final volume of 25 µL) using primers oIMR6890 (Fw) 5′-GACGCCATCCACCTATGAG-3′, oIMR6891 5′-GGCTGCTCCCATTCCACT-3′ (Rev), and oIMR6892 5′-GGCTCTTTGTGGGACCATCAG-3′ (Rev) accordingly to the B6.129-*St3gal4*^tm1.1Jxm^/J genotyping protocol [35]. The PCR program was 2 min at 95 °C, followed by 34 cycles of 95 °C for 30 s, 66 °C for 1 min, and 72 °C for 40 s. A final extension step was performed at 72 °C for 5 min. PCR reactions were kept at 4 °C until being electrophoresed in 2% agarose 1× TBE (89 mM Tris base, 89 mM boric acid, and 2 mM ethylenediaminetetraacetic acid) gels for 100 min at 85 V. For visualization of electrophoresed PCR products, the gels were stained with ethidium bromide (0.006% *v*/*v*) and digital images were captured in a CHEMIDOC MP Imaging System (Bio-Rad, Hercules, CA, USA). Homozygous WT, HZ, and KO genotypes were distinguished by their different band patterns on gel, as reported in relative genotyping protocol (290 bp, 290 bp and 450 bp, 450 bp, respectively).

### 2.3. Cross-Fostering and Weaning Procedure

Twenty WT and 20 KO female mice were mated with 10 WT and 10 KO male mice, respectively. Out of this batch, 16 WT and 16 KO dams gave birth to viable offspring. The fostering procedure performed between 10:00 a.m. and 1:00 p.m., required the use of at least four dams (two WT and two KO; Figure 1). To minimize the number of animals to be discarded due to the absence of foster dams, fostering procedures were performed between PND 1–2.5. On the day of fostering, we first removed the dams from their cage and then sexed and marked the offspring through toe tattoo ink puncture (Ketchum Manufacturing Inc., Brockville, ON, Canada) [36]. After sexing and marking procedures were completed, the pups were covered with sawdust and dams were relocated to their home cage. Each offspring was transferred to a foster dam to standardize fostering procedures across all experimental subjects. Each dam nurtured a mixed litter composed of WT and KO male and female offspring (1:1 ratio among all variables whenever possible). Out of the 32 experimental litters, it was not possible to fully balance all the aforementioned variables only in two litters. These two litters were kept in the study.

At weaning (PND 28), male mice reared to the same dam were marked through ear clippings, transferred together (two or three mice per cage) into standard type-1 polycarbonate cages (33.0 cm × 13.0 cm × 14.0 cm) and kept in the same conditions as described above. These procedures resulted in four experimental groups of male mice: CTRL (in-fostered WT offspring, *N*  =  18), consisting of WT mice receiving milk with 3′SL; MILK (cross-fostered WT offspring, *N*  =  14), consisting of WT mice receiving milk with reduced 3′SL level; GENE (cross-fostered KO offspring, *N*  =  15), consisting of KO mice receiving milk with 3′SL; GENE  +  MILK (in-fostered KO offspring, *N*  =  17), consisting of KO mice receiving milk with reduced 3′SL level. Developing offspring were subdivided into two cohorts. One cohort (CTRL, *N*  =  13; MILK, *N* = 12; GENE, *N* =13; GENE + MILK, *N* = 12) was tested for cognitive capabilities (attention and memory performances) and metabolic responses (glucose tolerance), and the other cohort (CTRL, *N*  =  5; MILK, *N* = 2; GENE, *N* = 2; GENE + MILK, *N* = 5) was used to assess electrophysiological correlates of memory performance (LTP in hippocampal slices, see test battery in Figure 2). To avoid litter effects, each group in each cohort consisted of mice born to different dams.

### 2.4. Behavioral Assessment

The following behavioral assessment was performed in an air-conditioned room (temperature 21 ± 1 °C and relative humidity 60 ± 10%) adjacent to the housing room: NOR, elevated 0-maze, PPI, Barnes maze, T-maze, ASST, glucose tolerance. The sucrose preference test and the assessment of general locomotion were performed directly in their home cages in the housing room. The Barnes maze test was performed under bright light, all other tests under dim light. With respect to behavioral outcomes, NOR, elevated 0-maze, PPI, Barnes maze, and general locomotion were quantified by automated software (“The Observer XT,” Noldus, Wageningen, the Netherlands, for NOR and elevated 0-maze; and “The Ethovision,” Noldus, Wageningen, the Netherlands, for Barnes maze), which, by definition, is blind to treatments; for T-maze and glucose tolerance test, the experimenter conducting the test received the mouse from another experimenter who guaranteed blinding; for ASST and sucrose preference, test blinding was not possible. Blinding to treatments in electrophysiology experiments was ascertained by providing the experimenters with samples that were labeled with codes masking the identity of the subjects.

### 2.5. Novel Object Recognition (NOR, Week 11)

The NOR paradigm was performed as described previously [37]. In brief, on the first day, the mice were individually habituated for 30 min to an opaque dark arena (40 cm × 40 cm × 40 cm; Technosmart Europe srl, Rome, Italy) equipped with a camera (Sony Handycam DCR-SX21E, Tokyo, Japan) under indirect dim light. On the second day, mice were exposed for 10 min to two identical unfamiliar objects made of plastic (A) located in direct contact with two opposite walls of the arena. To assess novel object recognition memory, each mouse was tested in two test trials conducted 1 h (short-term memory) and 24 h (long-term memory) after habituation. In these tests, the mice had to discriminate between one exemplar of the same object type (A) and one exemplar of another object type (B) (short-term memory) or (C) (long-term memory). An exploration ratio, calculated as the time spent exploring the novel object divided by the time exploring both objects, was used to measure object recognition memory. A camera mounted above the arena recorded each trial and videos were analyzed offline by a human operator (intrarater reliability coefficient 0.98). Exploration of an object was defined as directing the nose toward an object at a distance of less than 1 cm and/or touching the object with the nose and/or paws. Sitting on the objects was not considered exploratory behavior.

### 2.6. T-Maze Spontaneous Alternation Test (Week 12)

The animals were screened for perseverative behaviors in the T-maze test (see [32] for details). The experimental apparatus consisted of a closed T-shaped maze, composed of three equally sized arms (14.5 cm × 8 cm; Technosmart Europe srl, Rome, Italy), in which the mice were tested on 10 sessions to ascertain their spontaneous tendency to alternate between consecutive left–right binary choices [32]. We measured the spontaneous alternation index expressed as the number of alternations divided by the total number of sessions × 100. 

### 2.7. Elevated 0-Maze (Week 13)

To evaluate anxiety-related behavior, the mice were tested on the elevated 0-maze [38]. The apparatus was constituted by a circular runaway (5.5 cm wide) made in black plastic, with a 46 cm diameter, elevated 40 cm above the floor (Technosmart Europe srl, Rome, Italy), divided into four sectors: two of them, opposite to each other, were protected by 16 cm high walls made of transparent Plexiglas (closed sectors); the other two sectors were not protected by walls (open sectors). The experiment was recorded with a video camera (Sony Handycam DCR-SX21E, Tokyo, Japan). 

### 2.8. Pre-Pulse Inhibition (Week 14)

The apparatus (Med Associates Inc., St Albans, VT, USA), consisting of an acoustic stimulator (ANL-925, Med Associates Inc., St Albans, VT, USA) and a platform with a transducer amplifier (PHM-250-60, Med Associates Inc., St Albans, VT, USA), was positioned in a foam-lined isolation chamber (ENV-018S, Med Associates Inc., St Albans, VT, USA), defined as startle chamber. Experimental data were acquired automatically through dedicated software (SOF-815, Med Associates Inc., St Albans, VT, USA). The procedure adopted in this study, consisting of a habituation phase followed by a test phase on the following day, has been detailed in [32]. Pre-pulse inhibition was computed as the percentage of reduction of startle to the pre-pulse + pulse trials compared to the pulse alone trial 1 − (startle_pre-pulse+pulse_/startle_pulse alone_).

### 2.9. Barnes Maze (Weeks 15–16)

In this task, originally developed by Barnes [39], mice exposed to a bright light (85 lux) were required to locate a rectangular escape box (7 cm × 37 cm × 9 cm) located underneath one of 20 holes (target hole, diameter 5 cm; Technosmart Europe srl, Rome, Italy); the other 19 holes were covered with a black cap that provided the same visual cue offered by the target hole [39]. The circular arena, elevated 93 cm above the floor, had a diameter of 92 cm; the 20 holes were evenly spaced on the perimeter of the arena. This test entails 10 acquisition trials conducted on five consecutive days and two probe trials conducted 24 h and seven days after the last acquisition trial. Details on this procedure have been specified in [32]. Probe trials consisted of a 90 s free exploration during which the escape box was removed, and all the holes, including the target, were covered with the caps. Memory, during the probe test, was evaluated through the measurement of the time spent in the target zone. 

### 2.10. Attentional Set-Shifting Task (Weeks 17–20)

We adopted the attentional set-shifting task, originally developed by Birrell and Brown and modified by Colacicco and colleagues ([40,41]; see also [42]). The apparatus was composed of an opaque PVC U-shaped box with a grid floor with a transparent plexiglass lid (45 cm × 30 cm × 15 cm, home-made apparatus). Two identical compartments (15 cm × 15 cm) at one end of the apparatus could be accessed through doors from the starting compartment (30 cm × 30 cm). A round food cup (4 cm diameter, 3.5 cm high) in each choice compartment was baited with a small piece of cereal (30 mg; Honey Nut Loop, Kellogg, Battle Creek, MI, USA). The food was buried under a layer of scented digging medium (2 cm; Table 1). The food reward presence or absence was indicated by either tactile (digging medium) or olfactory stimuli (scent). On the day before testing, the mice were habituated to the apparatus for 10 min. Following this habituation, the mice were trained for a series of nine trials to dig into food-baited bowls. During the initial three trials, the mice were allowed to explore the apparatus until two food rewards located on the surface of the empty bowls were retrieved. During trials 3–6, the mice were allowed to explore the apparatus to collect two food rewards located on the surface of the digging media inside the bowls. In the final three trials, food rewards were located underneath the digging media. This procedure was put in place to ensure that the mice were able to perform reliable digging. During testing, trials were initiated by giving the mice access to the two digging bowls, only one of which was baited. The mice were food-restricted (90% of their body weight at the beginning of the experiment) and were required to dig into the rewarded bowl to obtain a highly palatable reward. Digging bowls varied across digging media and odors (see Table 1 for rewarded stimulus information). During simple discrimination (SD), the mice had to discriminate between two odors. The mice were then required to perform a compound discrimination (CD), during which the rewarded stimulus of the previous stage was presented together with a new stimulus of the other dimension (digging medium). In CD, the correct and incorrect exemplars remained constant (e.g., cinnamon odor was rewarded when combined with either sawdust or shredded paper while thyme was never rewarded, see Table 1). CD was followed by CD reversal learning (CDR). For the CDR, the mouse had to learn that the previously correct stimulus was now incorrect. In the intra-dimensional shift (IDS), we changed all the stimuli, but odor remained the relevant one, while in the extra-dimensional shift (EDS), the digging medium became the stimulus associated with the food rewarded. During the first four trials of each stage (SD to EDS), the door remained open even if the choice was incorrect, and the mouse was allowed to explore and collect the reward from the opposite bowl. In all trials, an error was recorded when the subject started to dig in the unbaited bowl. A stage was considered complete when the mouse achieved 8/10 correct trials. A session would continue until the animal stopped working. Normally, the mice would give a good response profile for about two hours. Since the end of a session depended on the individual subjects’ motivation, each subject underwent a variable number of daily trials. 

### 2.11. Sucrose Preference (Week 21)

To indirectly evaluate the integrity of the reward system, the mice were exposed to the sucrose preference test (see [43] for details on the procedure). This paradigm entailed a habituation phase (two days) and a test phase (five days). During habituation, the animals were exposed to only one bottle containing a 6% sucrose/water solution. During the test, the animals had access to two bottles filled with either water or a 6% sucrose/water solution, respectively; the two bottles were positioned randomly (left or right) on the lid of the animal cage and their position was randomly attributed every day. Fluid consumption was monitored every day for five days. During this test, the food positioned inside the home cage was available ad libitum. 

### 2.12. General Locomotion (Week 22)

Locomotor activity was evaluated continuously for four days in the housing room using infrared sensors positioned on top of the home cages. During this test, the mice were housed individually. An automated device using a small passive infrared sensor placed on top of each cage (ACTIVISCOPE system, NewBehaviour Inc., Zurich, Switzerland) was used [44,45]. Individual movement was detected with a sensor operating at 20 events per second frequency (20 Hz). 

### 2.13. Glucose Tolerance Test (Week 23)

Blood glucose concentrations were measured through a commercial glucometer (Accu-Chek Active, Roche Diagnostics) before and after an intraperitoneal (IP) injection of 2 g/kg body weight d-glucose (10% d-glucose solution; Sigma, St. Louis, MO, USA). Specifically, the mice were food-deprived for 15 h (18.30–9.30) and then injected IP with glucose. Blood glucose concentration was measured in baseline conditions (before the injection, t0), and then 20, 40, 60, 120 min after glucose injection. Peripheral blood samples were obtained from the central part of the tail, by the tail nick procedure [46], performed through a commercial razor for callus removers (SOLINGEN^®^, Solingen, Germany). To account for the integral response to glucose administration, we also calculated the area under the curve using the trapezoidal rule. 

### 2.14. Electrophysiology Experiments (Weeks 8–12)

LTP experiments were performed on hippocampal slices (*N* = 10 slices from five animals for CTRL, *N* = 4 slices from two animals for GENE, *N* = 4 slices from two animals for MILK, and *N* = 12 slices from five animals for GENE + MILK) collected in animals between 8 and 12 weeks of age, as substantially described in Martire et al. [47]. Briefly, the mice were sacrificed by cervical dislocation, and the brains isolated and immersed in ice-cold artificial cerebrospinal fluid (ACSF). The two brain hemispheres were separated and then sectioned by using a vibratome to obtain parasagittal slices (400 μm) containing the hippocampus. Extracellular field excitatory postsynaptic potentials (fEPSPs) were recorded in the stratum radiatum of the CA1 area after stimulation of Schaffer collaterals. Signals were acquired with a DAM-80 AC differential amplifier (WPI) and analyzed with the LTP program [48]. LTP was induced by a theta-burst stimulation (TBS) consisting of two trains of five sets of bursts (four stimuli, 100 Hz) with an interburst interval of 200 ms and a 20 s interval between each train. To allow for comparisons between different experiments, slope values were normalized, taking the average of the baseline values to be 100%. fEPSPs were recorded for 60 min after TBS and 10 min of stable baseline recordings preceded LTP induction. Changes in the fEPSP slope in the last 10 min of recording were expressed as percentage changes with respect to the average slope of the fEPSP measured during the 10 min that preceded the TBS. Curve fittings were obtained by using GraphPad Prism software (version 6.05, GraphPad Software, San Diego, CA, USA). 

### 2.15. Statistical Analyses

To investigate whether the experimental variables reflected independent aspects or aggregated underlying common factors, we preliminarily conducted a principal component analysis (PCA) on the 22 behavioral and physiological parameters (see Table 2). The PCA is a factorial method that allows correlation analysis of a set of *n* standardized variables by extracting k < *n* orthogonal factors as linear combinations of the original variables. These factors are then named after the domain that recapitulates the variables with the highest factor loadings (at least >0.5 in absolute value); this computation is based on the unrotated solution of the correlation matrix. After factor extraction, we adopted the scree plot procedure to select only those factors explaining more than 50% of the variance (Figure 3). Once the factors are defined, the loadings of each of the 22 variables are multiplied by the standardized values (for the z-score normalization, we used the following formula: z_i_ = (x_i_ − $\bar{x}$)/s, where z_i_ is the resulting z score, x_i_ the value to be normalized, $\bar{x}$ the sample mean, and s the sample standard deviation) and then added to identify the coordinates in a new multidimensional space. These new values (reflecting the score of each individual for each orthogonal factor) were then analyzed through ANOVA for split-plot designs. The general model entailed a 2 individual genotype (WT vs. KO) × 2 maternal genotype (WT vs. KO) statistical design. Individual and maternal genotype constituted between-subject factors. Fisher’s protected least-significance difference test was used for post hoc comparisons. Data are expressed as mean  ±  SEM. Statistical significance was set at *p* < 0.05. 

To provide a complete picture of the behavioral and physiological parameters investigated, we also analyzed all outcome measures independently. These analyses are reported in Appendix A. For behavioral and LTP data, statistical analyses were conducted using the software Statview 5.0 (Abacus Concepts, Berkeley, CA, USA). Data were analyzed through analysis of variance (ANOVA) for split-plot designs. The general model entailed a 2 individual genotype (WT vs. KO) × 2 maternal genotypes (WT vs. KO) × *k* (repeated measurements, variable depending on the specific test) statistical design. Offspring and maternal genotype constituted between-subject factors and repeated measurements constituted within-subject factors. Fisher’s protected least-significance difference (PLSD) test was used for post hoc comparisons. Data are expressed as mean ± SEM. Statistical significance was set at *p* < 0.05. Finally, in those instances in which a given threshold was required to confirm that experimental subjects met the criterion for a given experimental paradigm, the observed phenotype, reported as confidence interval (CI), was compared against the respective threshold through one-sample *t*-tests.

## 3. Results

### 3.1. Principal Component Analysis

The PCA extracted a total number of nine factors. To identify the smallest number of principal components explaining the largest proportion of variance, we observed that the first five factors explained 57.2% of the overall variance (Figure 3).

#### 3.1.1. Definition of the Independent Factors

While defining the principal components extracted, we observed that the first five factors were explained by the following variables (see Table 2 for the correlation matrix). Factor 1 (spatial memory and response to glucose) was primarily explained by the time spent in the target area of the Barnes maze both in the short and the long term, and by the immediate response to glucose injection (glucose response at *t* = 20). These variables are also associated with the distance moved in the Barnes maze apparatus during both probe trials. Finally, the general locomotion during the dark phase has high loadings on factor 1. Factor 2 (attention and glucose metabolism) was explained by the mid- to late-phase response to glucose administration and trials and errors in the attentional set-shifting task; importantly, these variables were inversely correlated and thereby a better performance in the ASST was associated with reduced reactivity to glucose administration. Factor 3 was named “working memory” as it was explained primarily by the spontaneous alternations in the T-maze and by the performance in the short-term memory probe of the Barnes maze (time in the target zone, latency to reach the target zone, and distance moved in the apparatus). Factor 4 (general locomotion) was characterized by the highest loadings for a single variable: locomotor activity during the dark phase. Finally, factor 5 (recognition memory) has elevated loadings for the time spent exploring an unfamiliar object in the NOR. We note that individual body weight had no significant loading on any of the principal factors extracted by the PCA. Body weight did not differ as a function of the experimental group (27.68 ± 0.44, 27.53 ± 0.39, 26.07 ± 0.57, 26.23 ± 0.68 for CTRL, MILK, GENE, and GENE + MILK, respectively; offspring genotype × maternal genotype: F_1,43_ = 1.57, NS).

#### 3.1.2. Spatial Memory and Response to Glucose

Reduced access to 3′SL during lactation resulted in reduced spatial memory (maternal genotype: F_1,39_ = 5.42, *p* = 0.025), whereby the MILK mice exhibited impairments in this factor compared to the CTRL subjects (Figure 4, Appendix A). We also observed that a genetic absence of 3′SL resulted in spatial memory impairments (offspring genotype: F_1,39_ = 24.28, *p* < 0.0001; Figure 4, Appendix A). Specifically, the GENE and GENE + MILK mice exhibited a decreased spatial memory performance compared to the CTRL subjects (Figure 4, Appendix A). The GENE + MILK mice exhibited further decreased performance compared to the MILK group. These observations are supported by statistical analyses conducted on the single variables contributing to factor 1 (see Appendix A for results on short- and long-term probe of the Barnes maze, glucose tolerance test, and general locomotion).

#### 3.1.3. Attention and Glucose Metabolism

The selective absence of 3′SL only during lactation resulted in deficits in attention and impaired glucose tolerance. Thus, the MILK mice exhibited reduced attention and a higher response to glucose injection compared to the three other groups (Figure 5). The effect seems to be moderated by the offspring genotype (offspring genotype × maternal genotype: F_1,39_ = 5.53, *p* = 0.02; *p* < 0.05 in post hoc tests; Figure 5, Appendix A). In the ASST, while the MILK mice required a higher number of trials and committed more errors than the CTRL group (*p* < 0.05 in post hoc tests), such difference was not present in KO offspring, with the exclusion of the EDS. In the glucose tolerance test at 40 and 60 min, the GENE + MILK group exhibited an increase in blood glucose concentration compared to the CTRL group, while both the MILK and GENE groups exhibited a decrease in blood glucose concentration, again supporting an interaction between offspring and maternal genotypes effects.

#### 3.1.4. Working Memory

The genetic KO of 3′SL had no significant effect on the working memory, although there was a trend to reduce working memory (offspring genotype: F_1,39_ = 3.21, *p* = 0.08) (Figure 6, Appendix A). The MILK mice did not differ from the CTRL subjects with respect to working memory by PCA (maternal genotype: F_1,39_ = 0.61, *p* = 0.26) (Figure 6). When working memory was assessed by using the T-maze spontaneous alteration test, we observed that the MILK mice were the only group exhibiting a reduced number of spontaneous alternations compared to the CTRL subjects (maternal genotype: F_1,41_ = 4.86, *p* = 0.03; see Appendix A).

#### 3.1.5. General Locomotion

When analyzed as a principal component, general locomotion did not differ between experimental groups (offspring genotype: F_1,39_ = 0.03, *p* = 0.86; maternal genotype: F_1,39_ = 0.001, *p* = 0.98; offspring genotype × maternal genotype: F_1,39_ = 0.34, *p* = 0.56; Figure 7). When analyzed as an independent parameter, experimental data suggested that the GENE + MILK mice exhibited lower levels of general locomotion compared to the other groups (Appendix A; Appendix A).

#### 3.1.6. NOR Memory

The absence of 3′SL during lactation resulted in NOR memory impairments, whereby the MILK mice showed lower values on this factor compared to the CTRL and GENE + MILK subjects by PCA (Figure 8). The recognition memory was not significantly different between CTRL and GENE or GENE + MILK groups by PCA (offspring genotype × maternal genotype: F_1,39_ = 4.84, *p* = 0.03; *p* < 0.05; Figure 8). The individual analysis on NOR discrimination ratio showed no statistically significant effect of MILK and/or GENE (Appendix A).

### 3.2. LTP

The reduction of 3′SL intake during lactation resulted in a higher level of LTP in acute hippocampal slices (CA1 area) of the WT mice cross-fostered to KO dams (interaction between offspring genotype and maternal genotype: F_1,10_ = 26.6, *p* = 0.0004; Figure 9a). The LTP magnitude of the KO pups cross-fostered to WT dams (GENE group) was higher compared to the hippocampal LTP of the KO pups fostered to KO dams (GENE + MILK group, see Figure 9b). Additionally, when compared to the CTRL subjects, the GENE mice exhibited higher levels of LTP (*p* < 0.05 in post hoc tests), while the GENE + MILK mice were indistinguishable (not significant, NS, in post hoc tests). Finally, compared to the MILK mice, while the GENE subjects showed an analogous level of LTP (NS in post hoc tests), the GENE + MILK subjects had a significantly lower level of LTP (*p* < 0.05 in post hoc tests, Figure 9b).

## 4. Discussion

In the present study, we observed that the selective deprivation of 3′SL during lactation resulted in an impairment of short- and long-term memory and attention. Specifically, we observed that the WT mice reared to KO dams, providing 3′SL-poor milk, exhibited reduced spatial and recognition memory compared to the WT mice reared to WT dams. This finding is based on the PCA analysis and confirmed by the individual statistical analysis conducted in the Barnes maze test. Additionally, although the PCA did not reach significance on working memory, specific analyses conducted on the T-maze indicated that the WT mice reared to KO dams had a suppressed working memory. These findings complement our previous study in which we obtained analogous results in response to the selective neonatal deprivation of 6′-sialyllactose (6′SL), another fundamental HMO [32]. These sugars (3′SL and 6′SL) constitute the most abundant source of sialic acid for the newborn, accounting for approximately 75% of breastmilk sialic acid content [49]. Therefore, the present findings further strengthen the view that an alteration of the main early-life dietary source of sialic acid can persistently influence the maturation of cognitive functions in mammals. 

The present results are in accordance with previous evidence reporting beneficial effects of dietary supplementation of sialic acid in piglets [30] and sialylated oligosaccharide in preterm pigs [50]. These previous studies were based on an experimental design that remarkably differed from the one adopted herein, whereby they supplemented source of sialic acid in a situation of physiological availability [30,50]. In Oliveros et al., maternal milk with a low level of sialic acid was obtained from the dams that delivered their pups 13 days earlier. Because the sialic acid component in breast milk reduces rapidly after 15 days, when the pre-weaned rats were cross-fostered to these dams, they naturally received a lower amount of sialic acid. By supplementing the feeding of these pups with sialic acid, the authors demonstrated the beneficial effect of sialic acid on memory functions [31]. This elegant design was nonetheless confounded by the fact that the breast milk with reduced sialic acid also had a different composition of other nutrients compared to the breast milk of their biological dams. Despite some differences in the protocols and experimental rationale, these studies support the role of sialic acid in the development of memory and cognitive functions in rodents. 

In addition, we evaluated LTP in hippocampal slices, trying to match our behavioral findings with a proper ex vivo experimental paradigm. This experiment, along with a previous study [32], further confirmed that the time-specific deprivation of sialic acid persistently alters LTP. It is important to emphasize that modulation of dietary sources of sialic acid resulting in increased LTP has also been observed in rats [30,31]. Notwithstanding independent evidence that LTP may be reduced as a function of altered neonatal concentrations of Sia [31,51], this multistrain, between-lab stability of findings confers a high degree of internal and external validity [52] to the hypothesis that sialic acid during lactation exerts a pivotal organizational role on LTP, a network function proposed to mediate memory. 

Besides memory impairments, in agreement with our previous study [32], the MILK mice also exhibited reduced attentional capabilities, one of the principal components of executive functions (EFs). These are defined as cognitive processes involved in planning and organizing behaviors through the monitoring of other low-level cognitive functions [53]. EFs are defined by three components: inhibitory control, working memory, and cognitive flexibility. The inhibitory control is necessary to selectively focus the attention on the purpose of the goal [54]. Working memory is essential to maintain the focus on the selected stimuli [55,56]. Finally, cognitive flexibility is built on the previous two components and is identified as the capability to rapidly switch between different perspectives and implement novel strategies to cope with adverse situations or changed circumstances [57]. The ASST used in the present study, requiring experimental subjects to disregard an acquired rule in favor of a new one upon the occurrence of mutated circumstances, directly addressed the behavioral flexibility domain of EFs. 

One last aspect that warrants particular attention relates to the potential role exerted by glucose metabolism in the cognitive impairments observed in our study. Specifically, we observed an inverse relationship between performance in the ASST and glucose metabolism, with higher blood glucose concentrations relating to increased errors to attain the criterion. Recent evidence, collected both in humans [58,59] and rodents [60], indicates that impaired glucose metabolism may predispose toward cognitive decline. Specifically, van de Vondervoort and collaborators [60] reported that an experimental model of type 2 diabetes (TALLYHO/JngJ mice), characterized by hyperglycemia, exhibited behavioral abnormalities analogous to those reported in the present study. Intriguingly, the association between hyperglycemia and altered cognitive capabilities has been demonstrated also in zebrafish [61]. Specifically, Ranjan and Sharma observed cognitive impairments in zebrafish immersed in a sucrose/water solution for 14 days. Ultimately, albeit preliminary, our data further suggest that somatic alterations may partly contribute to the onset of cognitive disturbances. 

Although the aim of this study was to evaluate the effect of a time-limited reduction of 3′SL on the development of behaviors related to executive function, our experimental design also allowed us to analyze the effect of the *St3Gal4* gene deletion. Experimental data showed that the GENE and GENE + MILK groups exhibited impaired short-term spatial memory capabilities compared to CTRL mice. In the GENE group, this impairment is associated with an alteration of the LTP parameters, while there were no differences in the other behavioral outputs, except that the GENE + MILK mice also exhibited a decreased locomotor activity. We suggest that this reduction in general locomotion is not likely to explain the differences observed in the other tests. Such proposition rests upon the following grounding: first, the main outcome parameters used to evaluate cognitive capabilities are not dependent on locomotion whereby they either rest upon binary choices not constrained in time (T-maze and ASST) or on percent preference values that, by definition, make data uniform irrespective of absolute levels of locomotion; second, had differences in locomotion explained the alterations reported in cognitive domains, we should have observed a consistent association between the former and the latter. Yet, while locomotion was reduced in the GENE + MILK mice, these mice were indistinguishable from the other groups in NOR and T-maze, different from CTRL but indistinguishable from the other groups in Barnes maze, and indistinguishable from the other groups in all stages of ASST apart from the IDS. Ultimately, the dissociation between general locomotion and the other parameters suggests that the former may not have exerted a major role on the latter.

Interestingly, the GENE and the GENE + MILK groups were indistinguishable compared to the CTRL subjects with respect to attentional capabilities. Regarding the GENE group, we hypothesize that the presence of 3′SL in milk potentially compensated for the constitutive knock-out of the *St3Gal4* gene. Concerning the GENE + MILK group, we observed a similar phenomenon in a previous study conducted on mice KO for the gene responsible for the synthesis of 6′SL (*St6Gal1*) [32]. In this study, we observed only minimal impairments in GENE (*St6Gal1*) + MILK mice and proposed a programming hypothesis [32]. Specifically, we suggested that in the GENE (*St6Gal1*) + MILK mice, the absence of the *St6Gal1* gene on the one side and the reduction of 6′SL in milk on the other side may result in the observed (lack of) phenotype. Specifically, while the absence of *St6Gal1* gene has been proposed to induce a reduction in host gut sialylation [62], 6′SL-poor milk has been associated with an alteration of the gut microbiota [63,64]. We hypothesize that a gut characterized by reduced sialylation is apt to interact with a microbiota grown in the absence of 6′SL in milk. Thus, a gut characterized by reduced sialylation may react negatively to a 6′SL-rich milk. We offer that analogous considerations may pertain to the findings obtained in the present study, conducted in mice KO for the *St3Gal4* gene. Albeit highly speculative, this hypothesis may also explain the LTP results wherein we observed remarkable alterations in the MILK and GENE groups but not in the GENE + MILK subjects. In-depth investigation of the impact on microbiota composition of these treatment would be necessary to assess whether this theoretical framework is valid.

## 5. Conclusions

In conclusion, by demonstrating that a selective neonatal deprivation of 3′SL results in long-term deficits in cognitive capabilities, this study further strengthens the developmental role of breast milk constituents. Specifically, this evidence, together with our previous findings on 6′SL [32], experimental data collected in other experimental species [30,31,50], and epidemiological data derived from formula-fed infants [13], highlight the importance of breast milk and breast milk constituents such as HMOs in brain and behavioral maturation.

## Figures and Tables

**Figure 1 nutrients-13-04191-f001:**
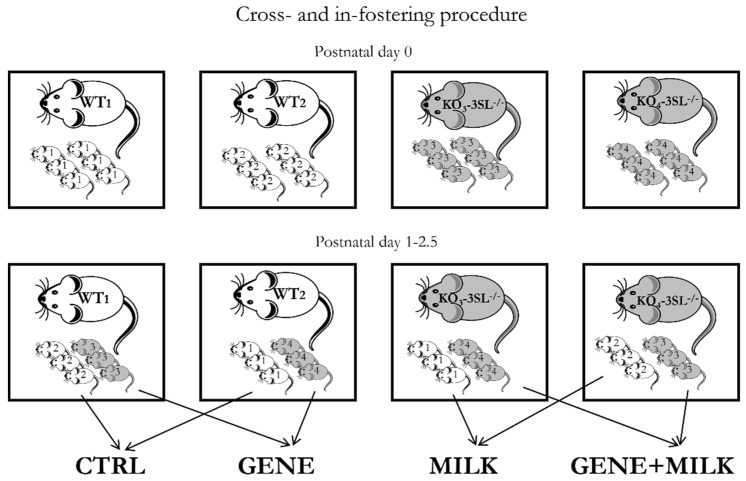
Fostering scheme. Cross- and in-fostering procedures were performed as shown in the lower panels between PND 1-2.5. Dams remained in their home cages while offspring were transferred from their original cages to those housing their foster dams. At the end of the fostering procedures, litters consisted of wild-type (WT) and knock-out (KO) mice in a 1:1 ratio. Importantly, no pup was reared to its biological dam.

**Figure 2 nutrients-13-04191-f002:**
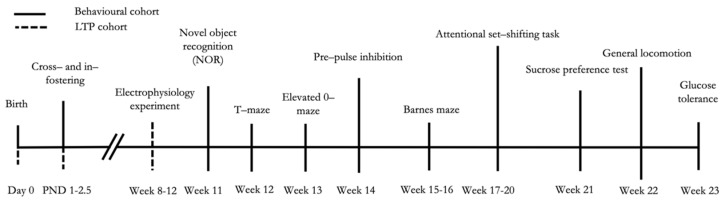
Experimental timeline. The day of birth was designated as postnatal day (PND) 0. The cross- and in-fostering procedures started between PND 1-2.5. After weaning (PND 28), the experimental subjects were divided into two cohorts: one exposed to a test battery entailing behavioral phenotyping and glucose tolerance test (CTRL, *N*  =  13; MILK, *N* = 12; GENE, *N* = 13; GENE + MILK, *N* = 12), and the other to electrophysiological assessment (long-term potentiation (LTP); CTRL, *N*  =  5; MILK, *N* = 2; GENE, *N* = 2; GENE + MILK, *N* = 5).

**Figure 3 nutrients-13-04191-f003:**
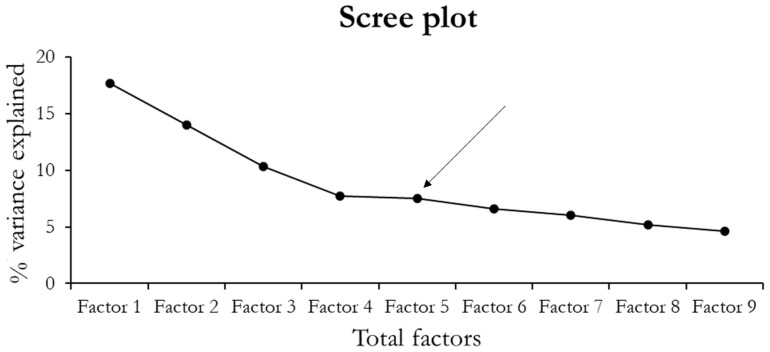
Scree plot. The figure represents the percentage of variance explained by the nine factors extracted during the PCA. The arrow indicates the elbow point that guides the selection of the factors for the ANOVA analysis.

**Figure 4 nutrients-13-04191-f004:**
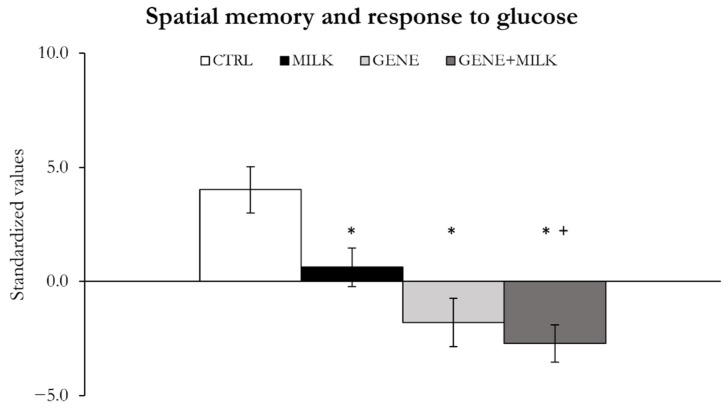
Spatial memory and response to glucose. Variable measured as mean ± SE of individual z-scores (sum of each standardized variable multiplied by its loading on factor 1). * *p* < 0.05 compared to CTRL, ^+^
*p* < 0.05 compared to MILK.

**Figure 5 nutrients-13-04191-f005:**
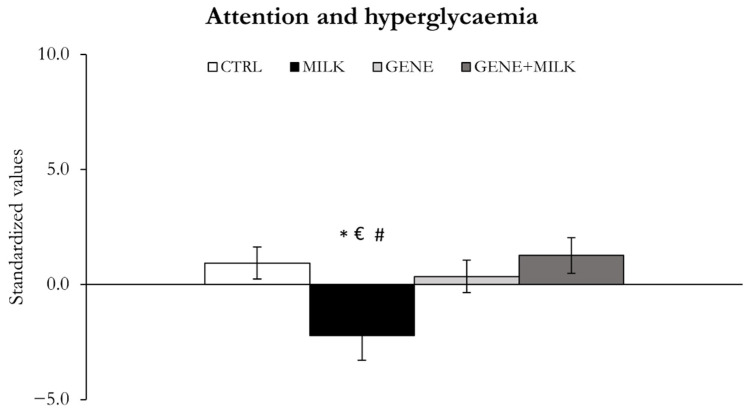
Attentional capabilities and hyperglycemia. Variables measured as mean ± SE of individual z-scores (sum of each standardized variable multiplied by its loading on factor 2). * *p* < 0.05 compared to CTRL, ^€^
*p* < 0.05 in post hoc tests compared to GENE group, ^#^
*p* < 0.05 in post hoc tests compared to the GENE + MILK group.

**Figure 6 nutrients-13-04191-f006:**
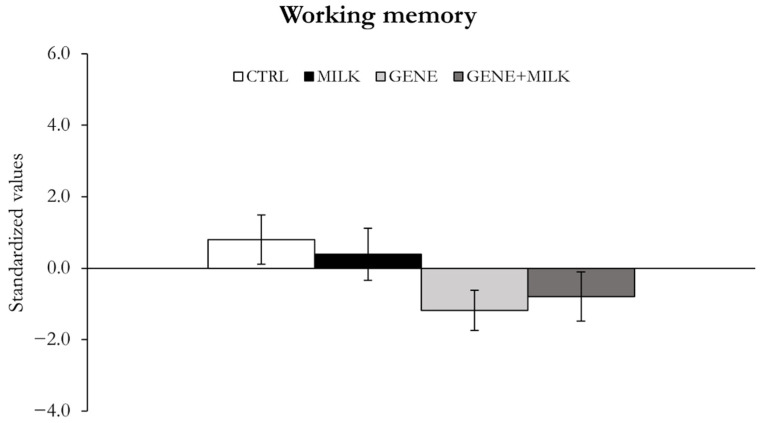
Working memory. Variables measured as mean ± SE of individual z-scores (sum of each standardized variable multiplied by its loading on factor 3).

**Figure 7 nutrients-13-04191-f007:**
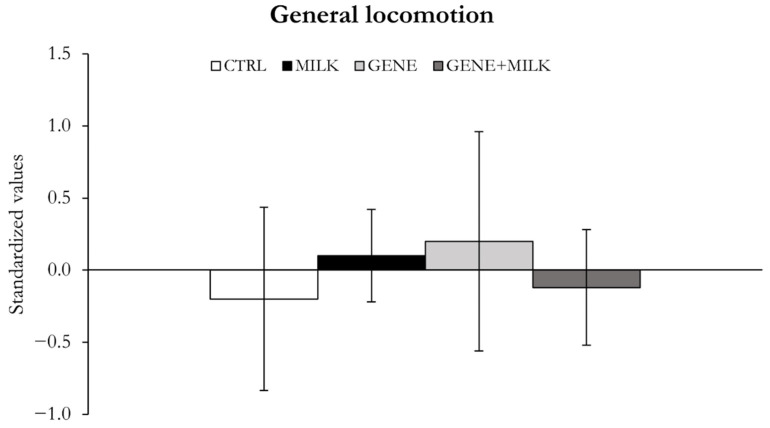
General locomotion. Variables measured as mean ± SE of individual z-scores (sum of each standardized variable multiplied by its loading on factor 4).

**Figure 8 nutrients-13-04191-f008:**
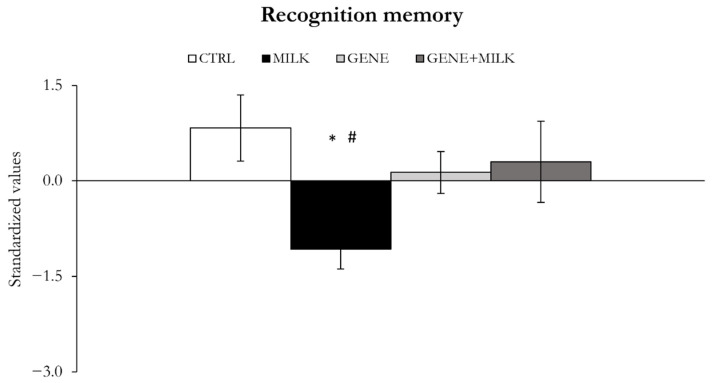
Recognition memory. Variables measured as mean ± SE of individual z-scores (sum of each standardized variable multiplied by its loading on factor 5). The MILK group recognition memory impairments compared to the CTRL group. * *p* < 0.05 compared to CTRL, ^#^
*p* < 0.05 in post hoc tests compared to the GENE + MILK group.

**Figure 9 nutrients-13-04191-f009:**
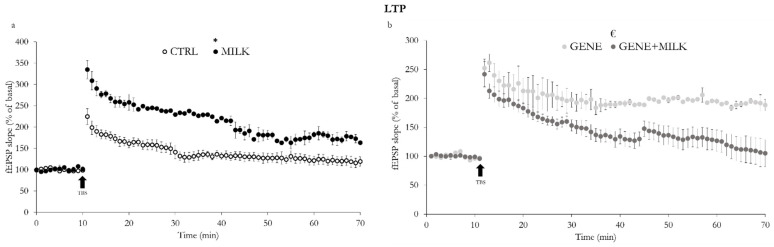
Extracellular field excitatory postsynaptic potentials (fEPSPs) recording (mean ± SE) in the CA1 area of hippocampal slices. (**a**), comparison between CTRL and MILK mice and (**b**), comparison between GENE and GENE+MILK mice: Long-term potentiation (LTP) was induced by theta-burst stimulation (TBS; indicated by the black arrow) of Schaffer collaterals and varied depending on the rearing dam (slices *N* = 10 for CTRL, *N* = 4 for MILK in (**a**) and *N* = 4 for GENE, *N* = 12 for GENE + MILK in **b**). * *p* < 0.05 compared to CTRL, ^€^
*p* < 0.05 in post hoc tests compared to the GENE group.

**Table 1 nutrients-13-04191-t001:** Stimulus examples used in the task.

Dimension	Pairing (Exemplar 1)	Pairing (Exemplar 2)
Odor	Cinnamon–thyme	Anuse–thyme
Medium	Sawdust–cotton	Sawdust–paper chip

Compound discriminations were based on fixed combinations of pairs of exemplars. The sequence of these combinations was presented in random combinations.

**Table 2 nutrients-13-04191-t002:** The unrotated factors extracted in the principal component analysis.

Variable	Factor 1 Spatial Memory and Response to Glucose	Factor 2 Attention and Glucose Metabolism	Factor 3 Working Memory	Factor 4 Locomotion	Factor 5 Recognition Memory
Alternations in the T-maze	0.111	0.179	**0.5**	0.324	−0.33
Preference for closed sectors in elevated 0-maze	−0.416	0.191	0.271	0.197	−0.431
Time in target zone in Barnes short-term probe	**0.663**	−0.377	**0.526**	−0.079	0.071
Latency to target in Barnes short-term probe	−0.258	−0.108	**−0.725**	0.189	−0.215
Distance moved in the Barnes short-term probe	**0.731**	−0.25	**0.504**	−0.063	0.087
Time in target zone in Barnes long-term probe	**0.512**	0.069	−0.08	0.465	0.185
Latency to target in Barnes long-term probe	0.046	−0.232	0.055	−0.401	0.355
Distance moved in the Barnes long-term probe	**0.645**	0.346	−0.242	0.399	0.122
Percentage of PPI	−0.132	0.236	0.492	−0.239	0.195
Total trials in ASST	−0.424	**0.677**	0.277	0.108	0.322
Total errors in ASST	−0.457	**0.672**	0.297	0.093	0.312
Sucrose preference	0.319	−0.272	−0.119	0.254	−0.227
Glucose tolerance *t* = 0	0.064	0.256	0.205	−0.343	−0.332
Glucose tolerance *t* = 20	**0.645**	0.366	−0.21	−0.143	−0.066
Glucose tolerance *t* = 40	0.483	**0.648**	−0.264	−0.251	−0.098
Glucose tolerance *t* = 60	0.325	**0.788**	−0.184	−0.062	0.092
Glucose tolerance *t* = 120	0.36	0.481	−0.143	−0.194	−0.056
Locomotion during dark active phase	0.023	−0.077	0.033	**0.611**	0.418
Locomotion during light inactive phase	**0.717**	0.032	0.174	0.102	−0.271
Preference for novel object short-term probe	0.244	−0.246	−0.131	−0.044	**0.629**
Preference for novel object long-term probe	0.134	−0.174	−0.091	−0.35	0.001
Weight	−0.047	0.081	0.366	0.276	−0.223

Table representing the unrotated factors extracted in the PCA. The parameters explaining each factor have been emboldened.

## Data Availability

The data presented in this study are available on request from the corresponding author. The data are not publicly available due to confidentiality agreement. Supporting data can be made available to researchers subject to a non-disclosure agreement.

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
