# Peer review of "Exposure to 3′Sialyllactose-Poor Milk during Lactation Impairs Cognitive Capabilities in Adulthood"

_nutrients, 2021, doi:10.3390/nu13124191_

Round 1

Reviewer 1 Report

In this manuscript, the authors investigated the effect of 3’sialyllactose-poor milk exposition on the cognitive capabilities of adult mice. The research topic is very interesting, experiments were conducted appropriately and the results are well discussed. I suggest to the authors some minor revisions:

  1. Consider specifying in the title that the study was conducted on mice.
  2. In the introduction, please add detailed information on the type and content of milk oligosaccharides, and in particular of sialylated milk oligosaccharides, in human milk and in the milk of traditional (cow, sheep, goat) and non-traditional (donkey, camel) dairy animals (see https://doi.org/10.3168/jds.2016-12388, https://doi.org/10.1017/S0007114513003772). Consider adding a short sentence on donkey milk, a promising source of sialylated oligosaccharides (see https://doi.org/10.1016/j.lwt.2019.108437).
  3. L161: Please add the age of breeding stock.
  4. L175 and L201: please standardize the style of writing numbers.
  5. L202-203: the sentence "Day of birth has been designated as PND 0" was already written at L171. Consider removing it from here.
  6. L270: check the bracket at the end of the line.
  7. The behavioural tests T-maze spontaneous alternation, Elevated 0-maze, Pre-pulse inhibition, Barnes maze and Sucrose preference were conducted according to? Please add the validated test references and consider reducing the text of these paragraphs. In addition, consider merging information on software and equipment once, when they are identical between the different behavioral tests.
  8. L654: breast milk refers to human milk, please modify.
  9. L664: delete the full stop at the beginning of the sentence.
  10. L698-699: please specify the animal model species.
  11. L702: add studies on another preclinical model that supports this hypothesis, such as zebrafish (see https://doi.org/10.1016/j.endmts.2020.100058).

Reviewer 2 Report

The authors submitted an article titled: “Exposure to 3’sialyllactose-poor milk during lactation impairs cognitive capabilities in adulthood”. They used a genetic model of the synthesis of sialyl(alpha2,3)lactose (3’SL) to examine the effects of 3’SL neonatal deprivation on a number of cognitive functions and synaptic plasticity marker (long-term potentiation). The paper gathers a lot of information in the field of cognition and its organisation and level of English are good. However, there are a lot of remarks to discuss that are numbered below and addressed to the authors:

  • Abstract, line 23: Please consider adding the word “neonatal” between “selective” and “lactational”.
  • Abstract, line 28: Please move “in adulthood” at the end of the sentence.
  • Introduction, line 40: please remove “individual”. Similarly in line 140.
  • Introduction, line 58: Please define breast milk administration period, as it is implied that adolescents received breast milk.
  • Introduction, line 60: Which are these “additional experimental evidence” that the authors refer to?
  • Introduction, line 73: Replace “for example” by “to this direction”.
  • Introduction, line 79: It is not clear why amygdala is referred here. What is the link to their study?
  • The introduction section is too long. Please consider reducing it to 3-4 paragraphs.
  • Methods: It is not clear why weaning occurred at postnatal day 28 and not 21, as usually conducted in other studies.
  • Methods, line 177: Replace “increasing” by “increased”.
  • Methods, line 179: The order of “0.5” and “0.3” is weird. Consider reversing them.
  • Methods, line 214: The 2 litters are excluded of the study? If not, why?
  • Figure 2: Consistency is needed e.g. days (PND) or weeks as time points. Also, for the NOR-object recognition memory. Please consider adding abbreviations in the legend.
  • Methods, line 251: Replace “were” by “was”.
  • Methods, line 257: Did the authors consider recording videos in order to achieve blind analysis?
  • Methods, Novel object recognition: It is not clear if the same mice were used for both 1h and 24h delays. Also, how was non-specific object exploration (e.g. when the mouse was close to the object without exploring it) excluded from the analysis? Was it verified by manual analysis?
  • Methods, Line 322: Correct “acclimation” to “acclimatization”.
  • Table 1: Consider adding “Table 1” in the legend.
  • Methods: It is not clear what happened to the brains of the mice in this study. Why not checking neurotrophic factors, e.g. brain-derived neurotrophic factor and its signalling?
  • The graphs of the study are not very comprehensible (e.g. regarding y axis or which test they refer to). Could the authors provide also the real values of behavioural performance of each test? Also, the error bars refer to standard deviation or to standard error?
  • Results, Line 611: no need of definition of CTRL.
  • Figure 9: In which brain area did the authors record the long-term potentiation?
  • Discussion: The structure of the first two paragraphs needs improvement.
  • Discussion, line 654: please remove “the” before “breast milk”.

Reviewer 3 Report

The article by Pisa et al describes the consequences of a 3’sialyllactose deficiency in milk on the cognitive abilities in adult male mice. The protocol involves the deletion of the enzyme responsible for the synthesis of 3’SL, and a cross fostering of the pups between WT and homologous deficient mice. The paper is very interesting and the behavioral study very complete and varied, with multiple tests. There are though a few points that need clarification or modifications.

  1. The introduction states that 3’SL and 6’SL are the most abundant HMO in milk. Here, only one enzyme was deleted. So have the authors inquired how much 6’SL was in the milk? Could this interfere in the deficiency of 3’SL? Could there be a compensatory mechanism?
  2. How did the pups grow? A few results about this point and adult body weights would be very much appreciated and are clearly lacking here. Also was the drive to suckle modified in the KO pups? Or was the milk intake similar between the different groups?
  3. Locomotion abilities were impaired in the group gene+milk, so how did this affect the other behavioral tests, since locomotion was involved in almost all the tests?
  4. It is a bit disappointing that the individual data are only under supplementary information. They are very important, and sometimes, more interesting that the PCA (e.g. lines 642 to 644). Lines 644 to 648 are confused and confusing. From line 714 to 728, it is not clear if the authors refer to 3’SL or 6’SL, since they refer to a published experiment about 6’SL. This whole paragraph should be clarified.
  5. Why only show the LTP from WT animals? The LTP from KO are interesting also.

The introduction is too long and should be shortened (lines 91 to 119 particularly).
